# Modularity through Attention: Efficient Training and Transfer of Language-Conditioned Policies for Robot Manipulation

**Yifan Zhou**
Arizona State University
yzhou298@asu.edu

**Shubham Sonawani**
Arizona State University
sdsonawa@asu.edu

**Mariano Phielipp**
Intel AI
mariano.j.phielipp@intel.com

**Simon Stepputtis**
Carnegie Mellon University
stepputtis@cmu.edu

**Heni Ben Amor**
Arizona State University
hbenamor@asu.edu

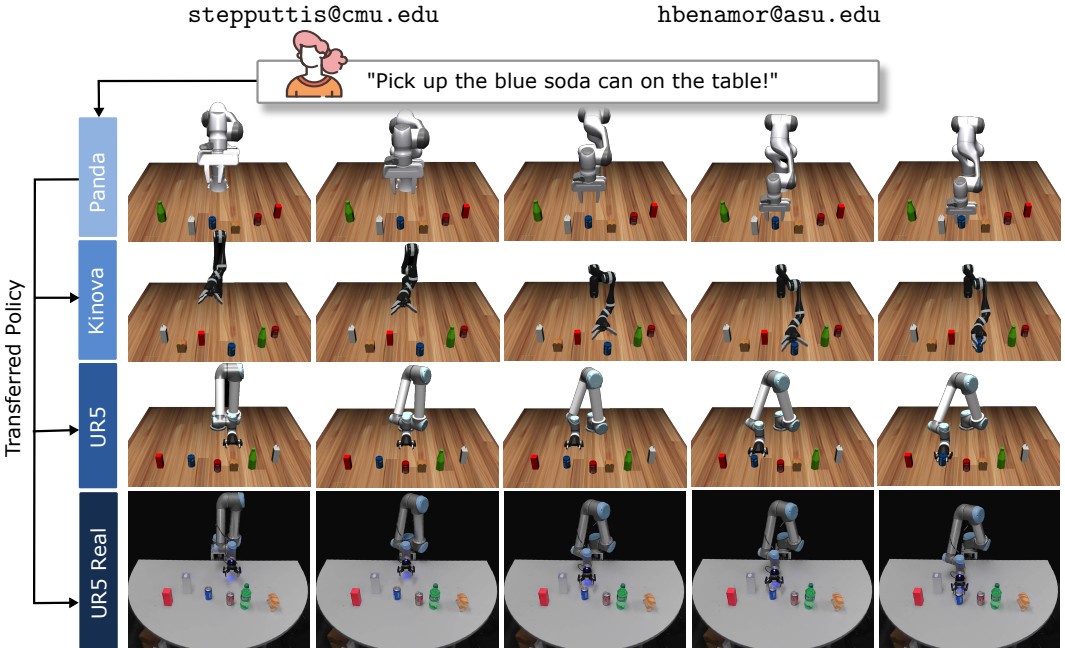

Figure 1: A human instruction is turned into robot actions via a learned language-conditioned policy. The neural network is then successfully transferred to different robots in simulation and real-world.

**Abstract:** Language-conditioned policies allow robots to interpret and execute human instructions. Learning such policies requires a substantial investment with regards to time and compute resources. Still, the resulting controllers are highly device-specific and cannot easily be transferred to a robot with different morphology, capability, appearance or dynamics. In this paper, we propose a sample-efficient approach for training language-conditioned manipulation policies that allows for rapid transfer across different types of robots. By introducing a novel method, namely Hierarchical Modularity, and adopting supervised attention across multiple sub-modules, we bridge the divide between modular and end-to-end learning and enable the reuse of functional building blocks. In both simulated and real world robot manipulation experiments, we demonstrate that our method outperforms the current state-of-the-art methods and can transfer policies across 4 different robots in a sample-efficient manner. Finally, we show that the functionality of learned sub-modules is maintained beyond the training process and can be used to introspect the robot decision-making process. Code is available at https://github.com/ir-lab/ModAttn.

**Keywords:** Language-Conditioned Learning, Attention, Imitation, Modularity.

6th Conference on Robot Learning (CoRL 2022), Auckland, New Zealand.

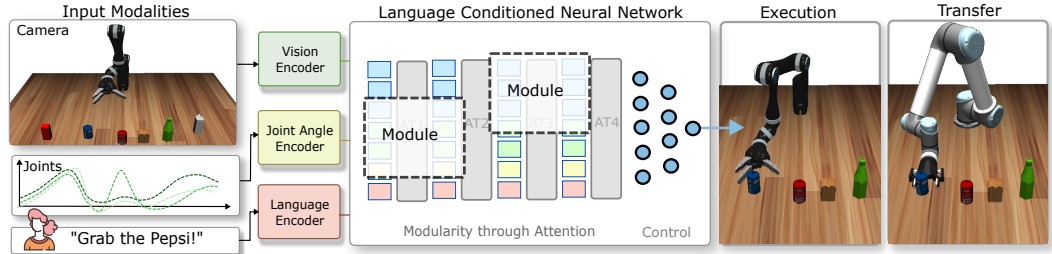

Figure 2: Overview: different input modalities, i.e., vision, joint angles and language are fed into a language-conditioned neural network to produce robot control values. The network is setup and trained in a modular fashion – individual modules address sub-aspects of the task. The neural network can efficiently be trained and transferred onto other robots and environments (e.g. Sim2Real).

# 1 Introduction

Creating machines that understand and physically execute natural language instructions is a long-standing vision of robotics and artificial intelligence [1, 2, 3, 4, 5]. To faithfully reflect the intentions of the human partner, robots need to interpret instructions within the current situational and behavioral context. For example, to execute the command *"Pick up the green bottle!"*, a robot may have to interrelate the words "bottle" and "green" to objects in the scene, identify the corresponding position in the image and, in turn, calculate the spatial relationship to its end-effector. Such inference and decision-making capabilities require a deep integration of multiple data modalities – inference at the intersection of vision, language and motion. Language-Conditioned Imitation Learning [2] addresses these challenges by jointly learning perception, language understanding, and control in an end-to-end fashion. However, once trained, such language-conditioned policies are not applicable to other robots. Due to the monolithic nature of end-to-end policies, robot-specific aspects of the task, e.g., kinematic structure or visual appearance, cannot be individually targeted and adjusted. Vanilla retraining on the new robot can possibly address this issue but comes with the risk of catastrophic forgetting [6] and substantial computational overhead.

In this paper, we address the problem of efficient training and transfer of language-conditioned policies across different robot manipulators. The overarching objective is to learn language-conditioned policies that can efficiently be transferred when the morphology, capability, appearance, or dynamics of the underlying robot changes. Instead of training an individual policy for every new robot, we aim to use methods that quickly transfer and adapt an existing master policy to new robot hardware, see Fig. 1 for an example. To that end, it is helpful to implement reusable building blocks or *modules* that realize specialized sub-tasks [7]. At time of transfer, robot-agnostic modules may not require any adjustments at all. However, such a modular approach is at odds with the monolithic nature of end-to-end deep learning. We therefore introduce a methodology which realizes such functionality through sub-modules that are responsible for achieving a specific target functionality. The approach builds upon two parts, namely *supervised attention* and *hierarchical modularity*. Supervised attention [8] was originally proposed for better alignment between two languages in the field of machine translation. In our approach, we leverage and extend supervised attention to enable the formation of modular components in networks. As a results, the user is able to manipulate the training process by focusing attention layers on certain input-output variables. By imposing a specific locus of attention, we guide individual sub-modules (or parts of attention layers) to realize an intended target functionality. Hierarchical modularity, on the other hand, is a training regime inspired by curriculum learning that aims at decomposing the overall learning process into individual subtasks. At the beginning of this process, only a single subtask is considered. After the corresponding module is trained to convergence, other tasks are addressed one-by-one within a hierarchical cascade. Combining both supervised attention as well as hierarchical modularity allows for neural networks to be trained in a structured fashion thereby maintaining a degree of modularity and compositionality. At runtime, the user can query both the overall control output of the network, as well as the outputs of each involved module. These queries can help to analyze the decision-making process by generating

the outputs of individual sub-networks. Misperforming modules can then be identified, isolated and retrained.

Our contributions can be summarized as follows: (1) we propose a sample-efficient approach for training language-conditioned manipulation policies that allows for rapid transfer across different types of robots, (2) we introduce a novel method, namely hierarchical modularity, which bridges the divide between modular and end-to-end learning and enables the reuse of functional building blocks, (3) we demonstrate that our method outperforms the current state-of-the-art methods (BC-Z [1] and LP [2]) and can transfer policies across 4 different robots (including from simulation to reality).

## 2 Related Work

Imitation learning aims at learning agent actions from expert demonstrations and has a rich history in robotics [9, 10, 11]. Previous works [12, 13, 14] have shown the feasibility of this approach for tasks such as helicopter flight, controlling humanoid robots, or collaborative assembly. However, early imitation learning methods are usually conditioned on low-dimensional sensor states and cannot leverage rich sensor modalities such as language and vision. Recent works have leveraged the representational power of deep learning to extended the paradigm to high dimensional language and vision understanding [15, 16, 17]. This progress is partly fueled by remarkable advances in the domain of image- and video-understanding [18, 19, 20, 21, 22]. For example, the work in [22] shows how vision and language can be projected into a joint embedding space. The result of these multimodal approaches enables a variety of downstream tasks, including image captioning [23, 24, 25], visual question answering systems (VQA) [26, 27], and multimodal dialog systems [28, 29]. A number of works leverage such multimodal networks as inputs to an autonomous agent [30, 31, 32, 33, 34, 35], thereby enabling the agents to understand semantics from raw inputs. Along this line of reasoning, language-conditioned imitation learning has been introduced to enable agents to act upon human verbal instructions [3, 2, 1, 5]. For example, BC-Z [1] proposes a large multimodal dataset which is trained via imitation learning. LanguagePolicies (LP) [2] uses a similar approach but describes the outputs of the policy in terms of a Dynamic Motor Primitive (DMP) [12]. More recently, Say-Can [4] was introduced which focuses on planning of longer horizon tasks and incorporates prompt engineering. Our approach follows a similar rationale to the above papers in that it aims at bridging the divide between language, vision and control through language-conditioned imitation. We do so by carefully incorporating modularity into our networks. Recent works on modularity for neural networks investigate, among other things, the question of whether "modules implementing specific functionality emerge" in neural network training [36]. Similarly, the work in [37] shows that pruning can produce surprisingly modular neural networks. Both of these works focus on a natural emergence of modularity, i.e., in a bottom-up fashion. Our work addresses modularity from a different vantage point. In particular, we ask the question, "Can we impose a specific set and order of modules within a regular neural network?". To this end, we introduce supervised attention within a hierarchical learning regime which allows for such functional modules to be implemented in a top-down manner. Supervised attention was initially introduced in machine translation by [8] for aligning two languages, and was proven effective in other natural language processing and visual question answering tasks [38, 39, 40, 41]. In this paper, we show that with the help of supervised attention, modular and reusable components can be formed in a hierarchical manner.

## 3 Language-Conditioned Policies for Robot Manipulation

The goal of our method is to learn a language-conditioned policy $\pi_{\boldsymbol{\theta}}(\boldsymbol{a}|\boldsymbol{s}, \boldsymbol{I})$ parameterized by $\boldsymbol{\theta}$, where $\boldsymbol{s}$ is the natural language sentence containing a human instruction and $\boldsymbol{I}$ is an image captured by an RGB camera of the entire scene. Policy $\pi_{\boldsymbol{\theta}}$ generates actions $\boldsymbol{a} = [x, y, z, r, p, y, g]^T \in \mathbb{R}^7$ where $(x, y, z)$ and $(r, p, y)$ are the task-space coordinates and orientations of the end-effector separately and $g$ is the gripper control signal. We train our policy by following the typical imitation learning paradigm by providing a data set of expert demonstrations $\mathcal{D} = \{\boldsymbol{d}_0, ..., \boldsymbol{d}_n\}$. Each demonstration $\boldsymbol{d}$ is a sequence of tuples $((\boldsymbol{a}_0, \boldsymbol{s}_0, \boldsymbol{I}_0), ..., (\boldsymbol{a}_T, \boldsymbol{s}_T, \boldsymbol{I}_T))$. After training, the system extracts a policy $\pi_{\boldsymbol{\theta}}$ which allows it to perform new configurations of the task.

An overview of our method is depicted in Fig. 2. A camera image, along with a natural language instruction and robot proprioceptive data (joint angles) is first processed through modality-specific encoders to produce embeddings. The resulting embeddings are passed as input tokens into a transformer-style [42] neural network with multiple attention layers. This neural network implements the overall policy $\pi_\theta$ and generates the robot controls. A crucial element of our approach is that the neural network is trained in a modular fashion, while still maintaining the advantages of end-to-end learning. The transition from training the components to training the overall network occurs gradually. Due to its modularity, $\pi_\theta$ can also be transferred to a new robot in a sample-efficient manner (e.g. from Kinova to UR5). Below, we describe the elements of our approach in more detail.

## 3.1 Modality-Specific Encoding

Given an image $I \in \mathbb{R}^{H \times W \times 3}$, an image encoder produces an embedding $e_I = f_\mathcal{V}(I)$. Inspired by [43, 44], we encode the input image into a sequence while keeping the original spatial structure. To this end, we first use a convolutional neural network to decrease its resolution but expand its channels, i.e., $e_I \in \mathbb{R}^{(H/s) \times (W/s) \times d}$, where $s$ is a scaling factor and $d$ is the embedding size. The resulting low-resolution pixel tokens are flattened into a sequence of tokens $e_I \in \mathbb{R}^{Z \times d}$, where $Z = (H \times W)/s^2$. The language module takes in natural language as instructions and generates a language embedding $e_s = f_\mathcal{L}(s) \in \mathbb{R}^{1 \times d}$, where $s = [w_0, w_1, ..., w_n]$ is a sentence composed of a sequence of words $w_i \in \mathcal{W}$ with $\mathcal{W}$ being the vocabulary. For computing the language embeddings, we refine the pretrained language model from CLIP [22]. During the training process, we provide a template of well-formed sentences. However, during testing, we admit any free-form verbal instruction including malformed sentences, typos, or bad grammar.

## 3.2 Supervised Attention

Modern neural network architectures employ attention mechanisms [42] to enhance performance. In general, the input to an attention module consists of three sequences of tokens: queries $Q$, keys $K$ and a values $V$. The computational process underlying an attention layer can be expressed as:

$$O = \text{Attention}(Q, K, V) = \text{softmax}\left(\frac{QK^T}{\sqrt{d_k}}\right)V \tag{1}$$

where $d_k$ is the dimensionality of the keys. The output matrix of the above operation $O$, holds in every row $i$ the weighted sum of all value tokens. The weights correspond to the similarity between the $i$-th query token and the respective value tokens. The overall attention mechanism is typically learned in an end-to-end fashion without an explicit supervision signal for the attention layers. To better control the information flow during learning, we propose *supervised attention* – a specific optimization target for attention layers. The key idea underlying supervised attention is that information about optimal token pairings may be available to the user. If we know which key tokens are important for the queries to look at, we can treat their similarity score as a maximization goal. Fig. 3 shows example tokens generated from visual patches and the corresponding supervision

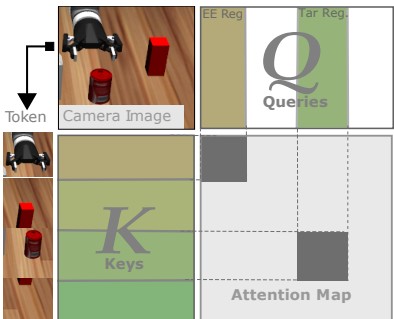

Figure 3: Supervised attention. Supervision pairs (dark gray) are provided by an expert to direct attention.

labels (dark gray). In the figure, we direct the network to focus on the end-effector and a target object. More formally, if the $i$-th query should look at the $j$-th key, then we need to maximize the similarity between $q_i$ and $k_j$, i.e., we optimize for $\arg\max_\theta\, q_i k_j^T$. This process is equivalent to maximizing the corresponding attention map element $M_{ij}$, where $M = \text{softmax}(\frac{QK^T}{\sqrt{d_k}})$. Note that, in contrast to matrix $O$ (Eq. 1), the calculation of the attention map does not involve the value matrix $V$. Since $M_{ij} < 1$, we can simply minimize the distance between $M_{ij}$ and

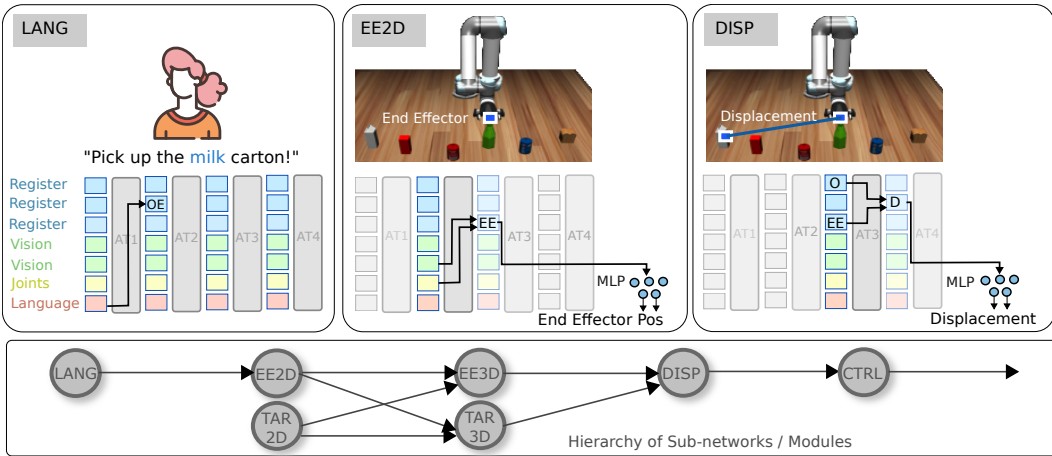

Figure 4: Hierarchical modularity: different sub-aspects of the tasks are implemented as modules (via supervised attention). LANG identifies the target object. EE2D locates the robot end-effector. The graph at the bottom of the figure shows the cascade of sub-tasks that are trained.

1. We assume that $N$ supervision pairs of indices for query and key tokens are provided, i.e., $\mathcal{S} = \{(i_0, j_0), (i_1, j_1), ..., (i_{N-1}, j_{N-1})\}$. Supervision pairs contain the indices defining which queries $\boldsymbol{q}_i$ should attend to which corresponding keys $\boldsymbol{k}_j$. Individual supervision pairs in this set can be addressed by $\mathcal{S}(p) = (i_p, j_p)$. Accordingly, we can define the following cost function for supervised attention:

$$\mathcal{L}(\mathcal{S}) = \sum_{n=0}^{N} \left( \text{softmax} \left( \frac{\boldsymbol{q}_r \boldsymbol{k}_s^T}{\sqrt{d_k}} \right) - 1 \right)^2 \tag{2}$$

where $(r, s)$ correspond to the indices held by the n-$th$ supervision pair $(r, s) = \mathcal{S}(n)$. The cost function in Eq. 2 casts supervised attention as a minimization problem, i.e., we calculate the mean squared error between the attention values and one. Other cost functions such as the cross-entropy [45] loss can also be employed but have yielded lower performance in practice.

### 3.3 Hierarchical Modularity

Using supervised attention, we can train attention layers that focus on specific input variables of interest. In turn, we can use this mechanism to train the attention layers to form sub-modules that realize target mechanism. The underlying rationale is to train part of the attention to realize components (or subtasks) of the overall policy. The sub-tasks, which are assigned to sub-modules embedded in the attention layers, are further arranged in a cascaded fashion, with the output token embeddings of one module being the input to the next module. Again, using supervised attention, the user can influence which variables (and which data modalities) are processed in a certain module.

**Algorithm 1** Hierachical Modularity: training algorithm returns network weights $\theta$.

**Input:** $\left( \mathcal{D}, \{\mathcal{S}_k\}_{k=1}^K, \{\mathcal{L}_k\}_{k=1}^K, \{\Psi_k\}_{k=1}^K \right)$
**Output:** Weights $\theta$
**for** subtask $k \leftarrow 1$ to $K$ **do**
    **while** not converged **do**
        $E_k \leftarrow \sum_{t=0}^{k} \mathcal{L}_t(\mathcal{S}_t) + \Psi_t$
        $\theta \leftarrow \text{Train}\left(\mathcal{D}, \{\mathcal{S}_1, \dots, \mathcal{S}_k\}, E_k\right)$
    **end while**
**end for**
**return** $\theta$

Inspired by Slot Attention [44], we also define register slots tokens (as shown in Fig.4), which are trainable tokens of embeddings that can propagate information throughout attention layers. For the purposes of robot grasping and manipulation, we identified the subtasks found in Tab. 1 which are implemented as individual modules within our framework. The first module LANG takes as input the sentence embedding of the human instruction and identifies the word embedding of the target object. Accordingly, when supervising the training of this module, we focus the attention on the language input. This process is visualized in Fig. 4. The figure shows a network of four attention layers, with the first layer implementing the LANG module. The output OE (object embedding) is turned

into an input token for the next layer. The TAR2D module takes an object embedding and generates an estimate of the object position in image space. Similarly, EE2D (Fig. 4) is trained to estimate the robot end-effector position based on information from the visual modality and the robot joint encoders. EE3D and TAR3D realize a similar functionality but focus on generating 3D positions in task space. A critical component is the DISP module, which estimates the displacement (or offset) of the robot end-effector from the target position. This information is crucial in order to enable closed-loop control. Finally, the output of the DISP module is used (together with all other input variable) to generate the robot control values CTRL. In our specific scenario, we predict the next 10 goal positions at every timestep, which are subsequently used by the operational space controller.

The overall learning process inherent to hierarchical modularization can be structured as a directed graph, see Fig. 4 (bottom). Every attention layer can implement one or multiple modules whose results are, in turn, fed to the next layer. The training process can be structured such that the elementary components necessary for complex decision-making are first derived within our cascaded modules, before being used for the final goal prediction. Algorithm 1 summarizes the the training process for hierachical modularity. The algorithm assumes that a set of attention

Table 1: Table of subtasks for modularity.

| Abbreviation | Subtask Description |
|---|---|
| **LANG** | Get target object from language |
| **TAR2D** | Find object patch |
| **TAR3D** | Calculate object position |
| **EE2D** | Find robot end-effector patch |
| **EE3D** | Calculate end-effector position |
| **DISP** | Find distance object to end-effector |
| **CTRL** | Predict robot positions for control |

losses $\mathcal{L}_k$, tasks-specific loss functions $\Psi_k$ and corresponding supervision signals $\mathcal{S}_k$ are provided by the expert for supervised training. The algorithm then starts training the first task using $\mathcal{L}_1$ and $\mathcal{S}_1$, which yields updated network weights. After convergence on this subtask, it appends the next cost function to the loss function $E$ and continues training the network. This process continues until all sub tasks are learned. Note that the final task, CTRL, is exactly the overall prediction target of our training process – the control signal for the robot. It is important to clarify two aspects underlying our methodology. First, we note that all modules are part of a single overarching neural network that implements the overall language-conditioned robot policy. Modularization is purely the result of training the network with different supervised attention targets and using a cost function that successively incorporates more and more sub-tasks. Second, the functionality of the modules is maintained even beyond the training process. Hence, at runtime, the user can query each of the modules (e.g. LANG, TAR2D, EE3D, etc) for their individual outputs.

## 4 Evaluation

We performed a number of experiments to show our model's ability to follow verbal instructions, its capability to transfer to new robots, and a real-world transfer from simulation. We also compare our approach to state-of-the-art methods for language-conditioned imitation learning, namely BC-Z [1] and LP [2]. We consider three types of manipulation actions to be performed by the robot: pick up a referenced object, push the object, rotate the object and put it down. There are six custom objects with different shapes and colors, which are a red cube, a Coke can, a Pepsi can, a milk carton, a green bottle, and a loaf of bread. All 3D models of these objects are from Robosuite [46]. **Evaluation Metrics:** Apart from recording success rates, we also evaluate the quality of each individual module within our language-conditioned policy, see Tab. 1. More specifically, we use as metrics: 1.) **Success Rate** describes the percentage of successfully executed trials among the test set, 2.) Target Object Position Error (**TAR3D**) provides the Euclidean 3D distance from the predicted target object position to ground truth, 3.) End Effector Position Error (**EE3D**) is the Euclidean 3D distance from the predicted end effector position to ground truth, 4.) Displacement Error (**DISP**) calculates the 3D distance between the predicted 3D displacement vector and ground truth vector.

### 4.1 Language-Conditioned Imitation Learning: Efficiency and Transfer

In this first set of experiments, we evaluated the ability of our approach to efficiently and accurately learn language-conditioned policies in simulation. Experiments were performed in MuJoCo [47]. We collected 2400 demonstrations, of which 1600 were used for training and 400 for validation

Table 2: Comparison with the state-of-the-art baseline as well as ablations, in Mujoco.

| Model | Success Rate (%) | | | | Prediction Error (cm) | | |
|---|---|---|---|---|---|---|---|
| | Pick | Push | Putdown | Overall | TAR3D | EE3D | DISP |
| BC-Z [1] | $81.6 \pm 6.2$ | $85.6 \pm 7.9$ | $49.1 \pm 5.8$ | $73.1 \pm 4.5$ | - | - | - |
| Ours (UR5 sim) | $\mathbf{91.3 \pm 5.3}$ | $\mathbf{97.2 \pm 2.0}$ | $\mathbf{55.6 \pm 8.6}$ | $\mathbf{82.4 \pm 4.9}$ | $\mathbf{2.24 \pm 0.48}$ | $0.51 \pm 0.09$ | $\mathbf{2.42 \pm 0.52}$ |
| Ours w/o Sup. Attn. | $88.9 \pm 4.2$ | $92.6 \pm 4.7$ | $55.1 \pm 11.6$ | $79.9 \pm 5.8$ | $3.18 \pm 1.80$ | $\mathbf{0.42 \pm 0.10}$ | $3.10 \pm 1.64$ |
| Ours w/o Hier. Mod. | $44.4 \pm 3.8$ | $39.4 \pm 7.1$ | $22.7 \pm 7.9$ | $36.4 \pm 3.3$ | $22.96 \pm 0.99$ | $0.59 \pm 0.16$ | $23.16 \pm 1.05$ |

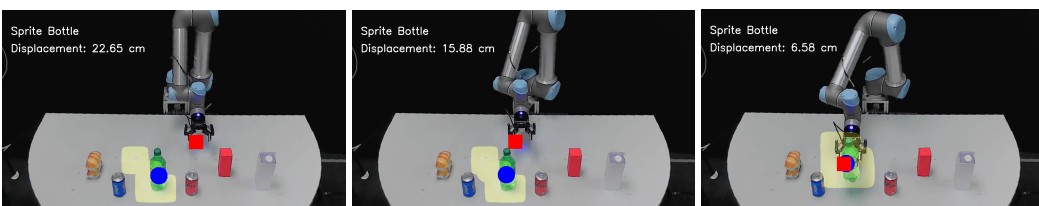

Figure 6: Sequence of real-time outputs of the network modules: the object name (white) and visual attention (yellow region), the length of the displacement (white text), the object pos (blue), and end-effector pos (red). All values generated from a single network that also produces robot controls.

and testing, respectively. Robot motions were generated using a simple motion planning technique towards the target object. Throughout robot execution, we recorded the natural language instruction (e.g., "Pick up the green bottle!"), a real-time stream of RGB images and robot proprioception (i.e., joints angles read from sensors). We trained LP using different hyperparameters eight times. However, we could not generate a performance better than random. This is expected performance since the size of training set is substantially smaller than LP's demand [2]. We trained and tested each method three times estimate the variability. Tab. 2, shows the result of this experiment.

As shown in Tab. 2, our approach achieves a high success rate ($> 90\%$) on both pick and push operations and outperforms BC-Z on all three tasks. The average success rate is 82.4%, compared to 73.1%. We also individually evaluated the modules EE3D, TAR3D, and DISP of the proposed network for its prediction error. We observed that the accuracy for the end-effector pose prediction ($\approx 0.5$cm) was higher than that of the target object, which is likely due to the availability of joint state data of the robot. The target object position can be approximated to about $2 - 3$cm, which is likely because no depth information is included in our input data (RGB only).

**Ablations:** We also tested the relevance of supervised attention and hierarchical modularity individually. Notice that in Tab. 2, removing hierarchical modularity causes a drastic drop in performance to a low of only 36.4%. In prediction error, we can see that the target and displacement error jump to over 20cm, explaining the low success rate. As noted before, the availability of information from the individual sub-networks increases the transparency of the model and our ability to introspect its behavior. Next, we removed the supervision signal on specific tokens; instead, attention is trained in an end-to-end fashion as typically done. Here too, the performance drops but only by about $2.5\%$. In Tab. 2, we see that the target object can still be identified via hierarchical modularity but generates higher prediction error and variance.

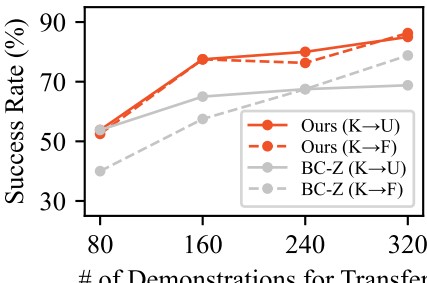

Figure 5: Results of transferring policies from Kinova (K) robot to UR5 (U) and Franka (F) robots. Experiments are performed in Mujoco simulator.

**Transfer:** The core hypothesis of our approach is that a modularly trained policy network will allow for better transfer to other robots with the minimal demand of retraining model parameters. We put this hypothesis to test, by using a single source model which was trained on a Kinova robot and

then transferred to Franka and UR5 robots in simulation. During transfer, we fine-tune our model on a new dataset recorded on the target robot (Franka or UR5). To evaluate data efficiency, we performed four tests with datasets of size 80, 160, 240 and 320 demonstrations respectively. Fig. 5 depicts the results of this analysis. We notice that our method achieves a success rate around $80\%$ with only 160 demonstrations. This is not a significant drop compared to the previous training results achieved on 1600 demonstrations (see Tab. 2). When using 320 demonstrations the results are on par with training on 1600 demonstrations. Further, transfer to UR5 with 160 demonstrations even outperforms BC-Z trained on 1600 demonstrations. For a better comparison, we also show the results of fine-tuning a BC-Z model (trained on 1600 demonstrations on Kinova and then refined with $80, \ldots, 320$) in Fig. 5. It is important to note that Kinova and UR5 are 6 degrees-of-freedom (DoF) robots, whereas Franka has 7DoF. They also have substantially different kinematic configurations and visual appearance. Transfer can effectively be performed with a limited data set to overcome the variations in appearance and kinematic configurations.

**Transfer to Real Robot:** We also tested the transfer of a network trained in simulation to a real-world robot (Sim2Real). We collected 340 demonstrations on a real-world UR5 robot, of which 260 were used for training and 80 for validation. After transfer using our approach, we tested the transferred models on the real world setup. We compared the results to a.) using the simulation model directly on the real robot and b.) continuous training, i.e., fine-tuning of the pre-trained model on the real-robot. Each model was then tested using 30 trials on the robot. Our approach achieves 80% success rate after transfer. The continuous training setup achieves a success rate of $56.67\%$; a difference greater than 20% compared to our proposed method. The model solely pre-trained in simulation (predictably) fails to perform any successful actions resulting in $0.0\%$ success. This effect can be explained by the substantial variation in visual appearance of objects and end-effectors in real-world images. We also observe the noise level change in the attention maps. Continuous training without supervising the attention results in a higher noise level in the attention map compared to adopting supervised attention.

**Extension of new modules:** We investigated the ability to add new modules to the existing hierarchy by adding an obstacle avoidance module. **Generalization on different scenarios:** We also tested the model's performance under varying environmental and task conditions, e.g., unseen colors, object scaling, object synonyms, image occlusions and unseen object types. **Linguistic variation:** Finally, we also asked random testers to paraphrase 30 natural language sentences for testing. Please see the appendix for details on the experiments mentioned.

## 5 Conclusions & Limitations

Our method demonstrates data-efficient training and multi-robot transfer of language-conditioned policies for robot manipulation. To this end, we introduce a novel method, namely Hierarchical Modularity, and adopt supervised attention to train a collection of reusable sub-modules. We also show that the learned hierarchy of sub-modules can be used to introspect and visualize the robot decision-making process.

**Limitations:** A major assumption made in our approach is that a human expert correctly identifies components and subtasks into which a task can be divided. This process requires organizing these subtasks into a hierarchical cascade. Early results indicate that an inadequate decomposition can hamper, rather than improve, learning. Further, our method can only work with instructions containing a single target object. Similarly, the approach does not incorporate memory and therefore cannot perform sequential actions. In a few cases we observed a failure to stop after finishing a manipulation - the robot continues with random actions. Regarding language understanding, there still exist a large number of limitations to the approach, e.g., it cannot identify objects by spatial references ("next to") or by function ("which contains"). Apart from them, the model does not generalize well on completely new objects and unseen geometric features. This is largely due to the fact that we trained our vision network from scratch on a small dataset, instead of using a pre-trained larger vision backbone.

**Acknowledgments**

This research was partially funded by grants NSF CNS 1932068 and IIS 1749783.

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
