# OpenReview forum: "Modularity through Attention: Efficient Training and Transfer of Language-Conditioned Policies for Robot Manipulation"
_robot-learning.org/CoRL/2022/Conference — CoRL 2022 Poster_

### Official Review · Reviewer_piKo · 2022-07-30

**Originality:** Good
**Technical Quality:** Good
**Clarity Of Presentation:** Good
**Impact:** 3

**Recommendation:**

Weak Accept: I recommend accepting the paper, but will not argue for my recommendation if the majority of other reviewers have a different opinion.

**Summary:**

In the paper authors propose a sample efficient approach for training language-conditioned manipulation policies that can be transferred across robots with different morphology, capability, appearance or dynamics. They introduce two complementary methods: supervised attention and hierarchical modularity. Supervised attention optimizes attention map by maximizing similarity between the queries and the keys. Hierarchical modularity decomposes the task of object manipulation to a set of reusable subtasks such as object detection, end effector detection, distance calculation. Authors demonstrate in both simulated and real world robot manipulation experiments that the method outperforms the current state-of-the-art methods and can transfer policies across different robots in a sample-efficient manner.


**Issues:**

The paper is clearly written and easy to read. Some important details are missing such as: which policy was used to collect the data, examples of introspection of the learned policy, how key-query pairs were generated for the training of the supervised attention mechanism.

**Quality Of The Limitations Section:**

Limitations are addressed clearly

**Reviewer Expertise:**

4: The reviewer is confident but not absolutely certain that the evaluation is correct

**Robotics Focus:**

Sufficient demonstration on hardware

**Strengths And Weaknesses:**

strengths:
* supervised attention mechanism
* decomposition to hierarchical modules which also help to introspect the robots policy

weaknesses:
* authors used supervised data to train the manipulation policy, so in that part novelty is limited
* collection of data used for supervised training of attention and manipulation should be described in more details.
* no examples of introspecting the robots policy

**Summary Of Recommendation:**

Describe in more details data collection process and provide examples of introspecting of the robots policy.

---

> ### Author Response · Authors · 2022-08-21
> **Response to Reviewer piKo**
>
> We thank the reviewer for their review and comments, particularly the positive feedback on the supervised attention and introspection opportunities given our approach. Thank you also for pointing out that the paper is clearly written and easy to read. We truly appreciate it!
>
> **We have attached a video (in zip) which is an integral part of our rebuttal. We would therefore greatly appreciate it, if you could please watch the video!**
>
> We address your comments below:
>
> **C1:** ***“Authors used supervised data to train the manipulation policy, so in that part novelty is limited.”***
>
> **A1:** Yes, since we investigate imitation learning, we collect supervised data for training – No RL. However, a critical aspect of our approach is that trained models can be reused on other robots with limited retraining. This makes supervision more economical in that it can be performed a single time on a master robot and then transferred to other robots in an efficient manner. Especially in the manipulation domain, many robot arms are very similar in structure (6 or 7DoF robot arms). But traditionally we had to completely retrain a learned policy when going from one robot to another, even though only certain aspects of the policy have changed. Being able to transfer policies between these robots would be of great benefit in many industrial tasks where interoperability of robots is in high demand. Our experiments credibly demonstrate that this is possible.
>
> **C2:** ***“Collection of data used for supervised training of attention and manipulation should be described in more details”***
>
> **A2:** Thank you for pointing this out! We have extended our description of the process in the new supplemental pdf, which you will find attached to this response (Sec. 2). We describe the setup and how query-key pairs are generated.
>
> **C3:** ***“no examples of introspecting the robots policy”***
>
> **A3:** Actually, Fig. 6 in the paper shows an example of the introspection process. The length of the displacement (DISP) is written in white, the object pos (TAR2D)  is displayed as a blue circle, and end- effector pos (EE3D) as a red box. A real time visualization of the introspection can also be seen in the attached video file (one visualization in simulation and one on the real robot). In addition, Tab. 2 in the paper shows the individual errors of each of the submodules of the learned hierarchy. For example, the module for detecting the end-effector position (EE3D) approximates the position to about 0.5 centimeters accuracy.

---

> ### Author Response · Authors · 2022-08-21
> **Supplementary Materials**
>
> **Comment:**
>
> * PDF explaining the details of additional information
> * Video showing supplementary experiments
>
> **Zip File:**
>
> /attachment/16ad540505203fd8cacbd91299045d4d8bffed4f.zip

---

> ### Author Response · Authors · 2022-08-28
> **Check back**
>
> ​​Dear Reviewer piKo,
>
> We would like to thank you for the constructive feedback on the paper! We have taken your feedback very seriously and have added important details regarding how data was collected etc. to the newly attached supplemental material.
>
> Regarding introspection: a real time visualization of the introspection can also be seen in the attached video file (one visualization in simulation and one on the real robot). We also performed a set of new experiments to investigate the generalization capabilities of the method. Our above responses also pinpoint where in the paper information about the introspection can be found. We have attached a video (in zip) which is an integral part of our rebuttal. We would therefore greatly appreciate it, if you could please watch the video!

---

### Official Review · Reviewer_fi3K · 2022-08-02

**Originality:** Very Good
**Technical Quality:** Very Good
**Clarity Of Presentation:** Excellent
**Impact:** 3

**Recommendation:**

Strong Accept: I recommend accepting the paper and will argue for my recommendation even if other reviewers hold a different opinion.

**Summary:**

This paper presents an approach for learning transferable language-conditioned pick & place policies by exploiting the inductive bias of attention to define chainable modules for performing tasks. Because there is a separation induced by the modules that are defined (e.g., identifying object to manipulate, estimating distance from end-effector to target object, mapping this distance to a control action), the proposed approach is able to show sample-efficient transfer to new robot morphologies, as well as across environments (sim2real).

In general, this paper proposes a neat, factored architecture for modular language-conditioned manipulation, and shows that adding such structure leads to better performance compared to other language-conditioned baselines, as well as meaningful transfer.

**Issues:**

Mostly, I’d love to see arguments addressing the point from above; furthermore, I’m curious as to why supervised attention, without the hierarchy is performing so much worse than BC-Z (Table 2). What is this approach missing that is present in BC-Z?

**Quality Of The Limitations Section:**

Limitations are addressed clearly

**Reviewer Expertise:**

5: The reviewer is absolutely certain that the evaluation is correct and very familiar with the relevant literature

**Robotics Focus:**

Sufficient demonstration on hardware

**Strengths And Weaknesses:**

In general, I like the punchline of this paper — by formulating a policy architecture with components that are human-interpretable (e.g., attention), we can leverage extra supervision to aid in sample-efficient learning. Going further, we can embed different “modules” with different functionalities within this architecture (as functions of different types of attention), and use this to guide task solving for a single robot, as well as sample-efficient transfer when generalizing to new robot morphologies and environments.

---

While the approach is straightforward, my key concern, and the biggest weakness of this paper, is the generality of the proposed modules. Just like Neural Module Networks in the computer vision & NLP communities (the end2end Neural Module networks paper should probably be cited here), the benefit of the predefined modules only exists as long as the task specification is relatively clean, and matches the hierarchy exactly — in this case, we’re mostly looking at object relocation without meaningful occlusion or other challenges, so the policy holds up.

However, where does the existing hierarchy break down? What if the type of objects changed (e.g., deformable, fragile, requiring different grasping policies)? What if there were meaningful occlusions or obstacles (requiring moving around existing objects in order to make the grasp we want)? While traditional BC approaches (e.g., in BC-Z) are able to scale to these new scenarios in the limit of new data, it’s not clear that this approach, with a fixed hierarchy, is able to scale similarly, which is really concerning! I’d love to see experiments looking at how the proposed hierarchy (e.g., this exact set of modules) performs in various settings, and whether there’s a clean way to add new elements to the hierarchy in an efficient, elegant way.

**Summary Of Recommendation:**

In general, while I really like the idea of supervised attention, I’m not sure that a single hierarchy is sufficient or general enough to tackle the messiness of real-world manipulation; I’d love to see new evaluations that show if/when the current hierarchy breaks down, and whether there is a clean way to fix such problems.

As a result, I advocate for a weak accept at this time.

EDIT (Post-Rebuttal): Changed score to Strong Accept!

---

> ### Author Response · Authors · 2022-08-21
> **Response to Reviewer fi3K (Part 1)**
>
>
> We would like to thank the reviewer for their valuable insights and for your enthusiasm regarding the paper, in particular that the paper presents “a neat, factored architecture for modular language-conditioned manipulation”. We are very glad to see that you enjoyed reading the paper!
>
> **We have attached a video (in zip) which is an integral part of our rebuttal. We would therefore greatly appreciate it, if you could please watch the video!**
>
> We performed a variety of additional experiments, as suggested, and address your comments below:
>
> **C1:** ***However, where does the existing hierarchy break down? What if the type of objects changed (e.g., deformable, fragile, requiring different grasping policies)? What if there were meaningful occlusions or obstacles (requiring moving around existing objects in order to make the grasp we want)? While traditional BC approaches (e.g., in BC-Z) are able to scale to these new scenarios in the limit of new data, it’s not clear that this approach, with a fixed hierarchy, is able to scale similarly, which is really concerning! I’d love to see experiments looking at how the proposed hierarchy (e.g., this exact set of modules) performs in various settings, and whether there’s a clean way to add new elements to the hierarchy in an efficient, elegant way.***
>
> Thank you very much for this suggestion! Yes, we are happy to report that we conducted all of the above experiments (and some more just in case).  Below experiments actually show that the approach scales on par, if not better, than traditional BC. **Please see the pdf attached to this response in Sec. 1 for more details on the experiments!**. Here a short summary of results:
>
> * An occlusion experiment: Objects were occluded up to 80% of the pixels that encompass the object. We actually found that for small occlusion rates (10-20%) we outperformed BC-Z regarding task success rate. For high occlusion rates we perform about the same.
>
> * Obstacle avoidance: we added obstacle avoidance as the reviewer suggested. For that, we added a new element that detects the obstacle and feeds in a specific fashion into the hierarchy. The resulting behavior can be seen in the new video. We achieve a success rate of 88% on the pushing task. We are particularly excited about the quick turn-around/addition times for the new element. Thanks to the modular approach, we were able to incorporate and train this new ability in less than two days.
>
> * Unseen object synonyms: we performed an experiment on the pushing task with 10 synonyms per object (all unseen during training). The object name was replaced with these synonyms during testing. This experiment shows that the underlying language understanding is not brittle and generalizes to different referrals to the same object. Success rate: ~82.5%.
>
> * Colors: We also noticed a remarkable generalization to changing colors on our objects - we can refer to an object with a description such as “fetch the purple object” and the network will correctly identify it even though purple was not part of the training set. Task execution is also robust to similar colors (e.g. red) being present. Success rate on pushing task: ~74.%.
>
> * Scaling: We also performed an experiment on pushing task in which objects are scaled by a random factor in one or more dimensions with up to 3x original size. Success rate: ~57.89%
>
> * New objects: The method is, however, not successful when completely new objects are introduced (success rate ~34.6% on pushing task). This is to be expected, since we did not train with a large object dataset. We argue that this was not the focus of the paper and good visual generalization can be achieved using widely known methods such as unsupervised pre-training, using vision backbones (VGG, ResNet, etc.). As shown in [r1], leveraging these pre-trained NN architectures is the key to visual generalization. We opted not to incorporate that to stay within the limits of our modest compute resources while still performing a number of ablations etc.
>
> * Unseen free-form instructions: we had already tested our method with new (unseen) free-form instructions for human users on all tasks, which resulted in an accuracy of 73.3% (see Line 295 in the original paper).
>
> [r1] Shridhar, Mohit, Lucas Manuelli, and Dieter Fox. "Cliport: What and where pathways for robotic manipulation." Conference on Robot Learning. PMLR, 2022.
>
> **C2:** ***“the end2end Neural Module networks paper should probably be cited here”***
>
> Yes, thank you we will make sure to include it!

---

> > ### Author Response · Authors · 2022-08-21
> > **Response to Reviewer fi3K (Part 2)**
> >
> >
> > **C3:** ***“I’m curious as to why supervised attention, without the hierarchy is performing so much worse than BC-Z (Table 2).”***
> >
> > This is an excellent question! Frankly, we were originally also surprised, but investigated this in more detail. In the experiment without hierarchical modularization, we immediately trained with a giant cost function that contains all sub tasks (end-effector localization, object location, displacement, control). Training was NOT done in a hierarchical, incremental fashion but rather everything at once. Accordingly, it becomes a large multi-objective optimization problem that is tricky to train – the result is often a suboptimal tradeoff between the terms of the cost function. Here, BC-Z fares better because it is not hampered by a complex multi-objective cost function, i.e., we only trained BC-Z with control error as cost. However, when we add hierarchical modularization to our method, training first optimizes the cost function of the first subtask to convergence (e.g. LANG task) and then incrementally appends more and more tasks. The result is something akin to an annealing strategy: we first learn a simple version of the task and then incrementally add new components. Ultimately, this strategy yields superior performance, while still maintaining the individual modules. A similar insight has been made by Bengio et al. [r2] when studying curriculum learning; see Fig. 5 in that paper (we will also cite).
> >
> > [r2] Bengio, Yoshua, et al. "Curriculum learning." Proceedings of the 26th annual international conference on machine learning. 2009.
> >
> > **C4:** ***“I’m not sure that a single hierarchy is sufficient or general enough to tackle the messiness of real-world manipulation”***
> >
> > The specific hierarchy in this paper is a result of our problem definition. Generally, our approach is about the ability to create variable hierarchies and reuse, replace or retrain elements selectively. We show in the obstacle avoidance example that the hierarchy can quickly be adapted. Please see the new attached pdf attached in this response in Sec. 1.6.

---

> > > ### Comment · Reviewer_fi3K · 2022-08-26
> > > **Official Rebuttal Response**
> > >
> > > Thanks so much to the authors for an excellent rebuttal! I love the added video, and think it solidifies my position of wanting to accept the paper - enough to champion this paper to the other reviewers who are leaning towards rejection (I will be changing my score to a Strong Accept)!
> > >
> > > The experiments re: where the existing hierarchy breaks down are really well done and well thought-out, and I find them really convincing; the turnaround time on the experiments stemming from the modularity of this approach are just added selling points! I'm still not convinced that there are tasks where it's easy to write down a predescribed hierarchy, but in terms of the heavy tail of robotics tasks that we just want to "work" - I think this hierarchy is natural and easy enough to achieve. I like this work a lot, and I really like what the authors have done with the rebuttal.

---

> > > > ### Author Response · Authors · 2022-08-27
> > > > **Thank you!**
> > > >
> > > > Thank you so much! Throughout the rebuttal, the reviewer's comments/suggestions have been very thoughtful and we are extremely grateful for the enthusiasm and the kind words expressed above. We would like to extend our deep appreciation to the reviewer for a positive rebuttal process that definitely made the paper better.

---

> ### Author Response · Authors · 2022-08-21
> **Supplementary Materials**
>
> **Comment:**
>
> * PDF explaining the details of additional information
> * Video showing supplementary experiments
>
> **Zip File:**
>
> /attachment/594014672d7ff13676dacd34b06ee6fd3243405d.zip

---

### Official Review · Reviewer_CVpY · 2022-08-05

**Originality:** Very Good
**Technical Quality:** Very Good
**Clarity Of Presentation:** Very Good
**Impact:** 3

**Recommendation:**

Weak Accept: I recommend accepting the paper, but will not argue for my recommendation if the majority of other reviewers have a different opinion.

**Summary:**

This paper introduces a transformer-based policy class from learning manipulation policies for physical robots from demonstration. It shows how providing explicit supervision to the attention layers in the transformers can force each module in the network to specialize in a specific type of reasoning. The paper presents a method to introduce these supervision tasks gradually over time, starting from lowest level tasks, to build up the modular reasoning capabilities of the model. The paper demonstrates how the resulting modular architectures are better adaptable to new manipulator types with few demonstrations, compared to policies trained with traditional end-to-end task losses only.

**Issues:**

Suggestions:
- Line 163: Not sure what “part of the attention” is. Consider rephrasing, perhaps more formally?
- Line 186: What are “all other input variable”? Can these be listed out or defined?
- Section 3.3 is perhaps a bit too informal and so hard to understand what’s really going on. E.g. What is the meaning of “target position”? Is the entire network simply inferring an end-effector goal in task space? If so, why does it need this many modules to do so?
- Algorithm 1 uses symbols that are undefined in the context in which the algorithm appears. It does not add much information.
- This work is highly related to a line of work in instruction-following that enforces modularity in end-to-end networks using auxiliary losses, and uses that modularity for sim2real transfer of a control policy on a drone navigation task [CITE], along with follow-up work on a household navigation task [CITE].


Questions:
- Line 233: Doesn’t the network predict target end-effector poses? Why is motion planning done towards a “target object”?
- How are supervision losses L_{k} obtained for each task from the demonstrations? How is supervised attention obtained in real-world experiments? Is it manually human-specified or automatically extracted from the demonstrations?
- Could you better explain the “w/o Hierarchical Modularity” baseline? Does this baseline only train on the task loss and not on the intermediate attention losses? Or does it train on all the same losses as the full model, but jointly all at once without gradually introducing more complex tasks once simpler tasks have converged.
- Is this method of transfer across robots better than e.g. predicting grasp points / end-effector keypoints, and running a motion planner to control the robots?
- Line 292: What are “register slots”? Were these defined anywhere in the paper? What kind of noise was added? Was this noise of a relative or absolute magnitude with respect to the distribution of values in these “register slots”?

Typos:
- Line 200: Typo “the the”
- Line 201: Typo “tasks-specific”
- Line 308: “three times estimate” -> “three times to estimate”.


**Quality Of The Limitations Section:**

Limitations are addressed clearly

**Reviewer Expertise:**

3: The reviewer is fairly confident that the evaluation is correct

**Robotics Focus:**

Sufficient demonstration on hardware

**Strengths And Weaknesses:**

Strengths:
- Interesting approach to constructing a robot policy. It’s nice that it’s a unified approach, where every module has the same architecture, and the distinctions between modules arise only from the objectives with which the modules are trained. This simplifies the design of robot policies.
- Convincing real-robot results on transferring policies between robots with different kinematic configurations. This is an important area of study, especially for deep learning based policies.

Weaknesses:
- The description of the method is perhaps too informal and sometimes vague about what are the inputs and outputs of the different modules, what types do these inputs and outputs take, and how is information being fed into these attention networks. Could also use more info on what are the labels for attention supervision and how these labels are obtained from demonstrations.
- Only slightly outperforms the baseline BC-Z, but with a seemingly much higher annotation burden.
- The “language-conditioned” aspect of the paper is somewhat weak. The main benefit of language over other task specification methods (e.g. joystick control, clicking to indicate navigation or manipulation goals) is ability to support free-form natural language instructions that include complex constraints, compositional task structures, temporal structures etc. A core challenge in language-conditioned policies is supporting flexible task specifications in a large enough task space to be interesting. Experiments should demonstrate at minimum generalization to unseen instructions combining seen primitives [1, 3, 6], but recently (and ideally) also unseen objects and primitives [2]. This work seems to uniquely focus on picking tasks, so language is used only to specify a target object. Presenting this work as doing referring expression grounding for goal object specification [4] would be more accurate. Even then, there are only 6 specific target objects, so the flexibility of referring expressions is limited compared to other works in referring expression grounding.
- The two contributions of “supervised attention” and “hierarchical modularity” are arguably overstated as contributions of this paper. “Supervised attention” has been studied multiple times in NLP research, starting with [5]. This paper may at most take credit for testing this idea on real robots. “Hierarchical modularity” simply refers to phasing in additional losses over time starting from low-level towards high-level reasoning. It’s roughly equivalent to the pre-training and joint fine-tuning paradigm that’s been commonly used in deep learning for years, and is also related to training policies with auxiliary losses, a common approach to encourage modularity in an end-to-end system [7, 8]. This paper can rightfully take credit for studying this approach (including with appropriate ablations) with attention-based networks in a robot manipulation task. This in itself is a meaningful contribution, but phrasing “hierarchical modularity” as an innovation might be overstating it.

Review references:
[1] Arumugam, Dilip, et al. "Grounding natural language instructions to semantic goal representations for abstraction and generalization." Autonomous Robots 43.2 (2019): 449-468.
[2] Shridhar, Mohit, Lucas Manuelli, and Dieter Fox. "Cliport: What and where pathways for robotic manipulation." Conference on Robot Learning. PMLR, 2022.
[3] Williams, Edward C., et al. "Learning to parse natural language to grounded reward functions with weak supervision." 2018 IEEE International Conference on Robotics and Automation (ICRA). IEEE, 2018.
[4] Shridhar, Mohit, Dixant Mittal, and David Hsu. "INGRESS: Interactive visual grounding of referring expressions." The International Journal of Robotics Research 39.2-3 (2020): 217-232.
[5] Liu, Lemao, et al. "Neural Machine Translation with Supervised Attention." Proceedings of COLING 2016, the 26th International Conference on Computational Linguistics: Technical Papers. 2016.
[6] Lynch, Corey, and Pierre Sermanet. "Grounding language in play." (2020).
[7] Hermann, Karl Moritz, et al. "Grounded language learning in a simulated 3d world." arXiv preprint arXiv:1706.06551 (2017).


**Summary Of Recommendation:**

The paper presents an interesting policy model, a new method (in robotics) to train the policy for a manipulation task from demonstrations, and convincing results on physical robot. A particularly good aspect is the ability to adapt the policy to different robot manipulators. It's great to see such multi-robot experiments that are often lacking.

That said, in it's current state the paper is a little vague on some important details, and slightly oversells certain aspects related to the innovations in the paper. I would like to see these writing and presentation issues are addressed.

---

> ### Author Response · Authors · 2022-08-21
> **Response to Reviewer CVpY (Part 1)**
>
>
> Thank you for your valuable feedback! We really appreciate you pointing out our “unified approach” which “simplifies the design of robot policies“ and the “convincing real-robot results on transferring policies between robots.”.
>
> **We have attached a video (in zip) which is an integral part of our rebuttal. We would therefore greatly appreciate it, if you could please watch the video!**
>
> We performed a variety of additional of experiments and address your comments below:
>
> **C1:** ***“The description of the method is perhaps too informal and sometimes vague about what are the inputs and outputs of the different modules, what types do these inputs and outputs take, and how is information being fed into these attention networks. Could also use more info on what are the labels for attention supervision and how these labels are obtained from demonstrations.”***
>
> **A1:** Thank you for pointing out this source of confusion, we will revise the description and appreciate the feedback! We will provide concise descriptions of the inputs and outputs for the final version of the paper. We have already added a section to our supplemental material that describes the labels and the process of obtaining them from demonstrations. Indeed, this was vague in the paper. **To clarify, the annotation process is automated**. However, for real-world experiments there is some human effort that needs to go towards robot-to-camera extrinsics calibration etc. which are common tasks in robotics. We had conservatively accounted for these as an additional overall overhead of about 1hr. In retrospect, that paragraph was clumsily written in the original paper. More information in the supplementary material (Sec. 2.3).
>
> **C2:** ***“Only slightly outperforms the baseline BC-Z, but with a seemingly much higher annotation burden.”***
>
> **A2:** We respectfully disagree with the reviewer. As mentioned before, annotation is automated. We outperform BC-Z by up to 20% in transfer tasks (Fig. 5). Even when no transfer is needed (Tab. 2) we achieve a 9.3% improvement over BC-Z. These are significant margins, especially considering our domain which involves the combination of vision, language and control, where a small error in any of these modalities leads to overall failure. In fact, as described in Sec. 4.1, another SotA method called LP ([2] NeurIPS 2020) failed to generate results better than random in any of our tasks(!) That shows the complexity of these tasks. For a good-faith and truthful evaluation, we had even reached out to the first author of that paper, who helped in ensuring best practices and fair comparison when using LP. Regarding reduction in training data: with less than 160 demonstrations, we achieve a better success rate than BC-Z with 320 demonstrations (Fig. 5). In our experiments, data needs have repeatedly been cut by more than half. Therefore, we stand by our results and argue that 9-20% increase in performance and a reduction of data needs by a half is a significant improvement over prior work!
>
> **C3_1:** ***“The “language-conditioned” aspect of the paper is somewhat weak. The main benefit of language over other task specification methods (e.g. joystick control, clicking to indicate navigation or manipulation goals) is ability to support free-form natural language instructions… Experiments should demonstrate at minimum generalization to unseen instructions combining seen primitives [1, 3, 6], but recently (and ideally) also unseen objects and primitives [2].“***
>
> Thank you for suggesting these additional experiments! We performed a new set of experiments (we also have a comment, see next comment section C3_2). **Please see the pdf attached to this response in Sec. 1 for more details on the experiments!**. Here a short summary of results:
>
> * An occlusion experiment: Objects were occluded up to 80% of the pixels that encompass the object. Performing experiments on all of pick, putdown and push tasks, we actually found that for small occlusion rates (10-20%) we outperformed BC-Z regarding task success rate! For high occlusion rates we perform about the same.
>
> * Unseen object synonyms: we performed an experiment on the pushing task with 10 synonyms per object (all unseen during training). The object name was replaced with these synonyms during testing. This experiment shows that the underlying language understanding is not brittle and generalizes to different referrals to the same object. Success rate: ~82.5%.
>
>
> * Colors: We also noticed a remarkable generalization to changing colors on our objects - we can refer to an object with a description such as “fetch the purple object” and the network will correctly identify it even though purple was not part of the training set. Task execution is also robust to similar colors (e.g. red) being present. Success rate on push task: ~74.%.
>
> ...

---

> > ### Author Response · Authors · 2022-08-21
> > **Response to Reviewer CVpY (Part 2)**
> >
> >
> > * Scaling: We also performed an experiment in which objects are scaled by a random factor in one or more dimensions with up to 3x original size. Success rate: ~57.89%
> >
> > * New objects: The method is, however, not successful when completely new objects are introduced (success rate ~34.6% on push task). This is to be expected, since we did not train with a large object dataset. We argue that this was not the focus of the paper and good visual generalization can be achieved using widely known methods such as unsupervised pre-training, using vision backbones (VGG, ResNet, etc.). As shown in [r1], leveraging these pre-trained NN architectures is the key to visual generalization. We opted not to incorporate that to stay within the limits of our modest compute resources while still performing a number of ablations etc.
> >
> > * Unseen free-form instructions: we had already tested our method with new (unseen) free-form instructions for human users on all 3 tasks, which resulted in an accuracy of 73.3% (see Line 295 in the original paper).
> >
> > * Obstacle avoidance: we added obstacle avoidance to show that new modules can easily be integrated. For that, we added a new element that detects the obstacle and feeds in a specific fashion into the hierarchy. The resulting behavior can be seen in the new video. We achieve a success rate of 88% on push task.
> >
> > [r1] Shridhar, Mohit, Lucas Manuelli, and Dieter Fox. "Cliport: What and where pathways for robotic manipulation." Conference on Robot Learning. PMLR, 2022.
> >
> > **C3_2:** ***“Experiments should demonstrate at minimum generalization to unseen instructions combining seen primitives [1, 3, 6]”***
> >
> > **A3:** We respectfully disagree with the premise and the description of the above papers regarding instructions combining seen primitives. Paper [6] does not perform combinations of primitives. As explicitly stated in the paper:
> >
> > *From [6]: “the operator can compose a new task “put object in trash” at test time, simply by breaking the task into two text instructions: 1) “pick up the object” and 2) “put the object in the trash”.”*
> >
> > Each instruction only maps a single primitive (trained behavior) in [6], and the user has to break it up into individual sentences/instructions. Papers [1,3] are not learning language-conditioned policies and also do not involve any image inputs. Rather, they turn freeform instructions into formal specifications that are treated as reward functions. In [1] a classification approach is used to map free-form sentences onto a discrete set of formal subtask/goal specifications. The specification is then treated as a reward function for “enabling optimal planning in stochastic environments.” Fundamentally, this is a planning approach, with the main focus being the classification process which generates formal specifications from free-form language. Even then, the possible instructions are similar to what is used in our paper in that they refer to an action and an object, e.g. “Stand in the blue room”, “Go by the red chair”, etc. To avoid any mischaracterization from our side, we contacted the authors of [1,3] who confirmed the above summary of their methods. In our experiments, we show that we are, similar to [2,6], capable of composing complex tasks from a sequence of instructions. E.g., we are capable of picking, placing (with rotation) and pushing an object in a sequential manner by providing a sequence of instructions (please see attached new video).
> >
> > **C4:** ***This work seems to uniquely focus on picking tasks, so language is used only to specify a target object. Presenting this work as doing referring expression grounding for goal object specification [4] would be more accurate.***
> >
> > **A4:**: Partially. The policy does more than that. It identifies which action to perform and produces in every time step end-effector positions and orientations and can therefore generate complex trajectories that include rotations of the wrist. This is critical for manipulation actions. In addition, the gripper state (degree of openness) is also controlled in every time step. As a result, pushing involves closing the gripper and nudging an object, whereas lifting requires opening and closing at right time, etc. For example, the putdown action also includes a rotation (the object is rotated sideways before being put down). We argue that this allows for complex manipulations rather than being limited to only picking or general point to point motion. If we were to offload the motion to a path planner as suggested by the reviewer, we would have to repeatedly provide intermediate way-points to realize the above behavior which would defeat the purpose of using a planner since the planner would only generate short motion segments. ...

---

> > > ### Author Response · Authors · 2022-08-21
> > > **Response to Reviewer CVpY (Part 3)**
> > >
> > > Similarly, this would require creating separate actions for closing and opening of the gripper and would not allow for gradual, continuous changing of gripper opening while also moving the arm. There are a number of other unintended consequences of that choice (i.e. conflicts between planner and policy etc.).
> > >
> > > **C5:** ***“The two contributions of “supervised attention” and “hierarchical modularity” are arguably overstated as contributions of this paper… This paper can rightfully take credit for studying this approach (including with appropriate ablations) with attention-based networks in a robot manipulation task. This in itself is a meaningful contribution, but phrasing “hierarchical modularity” as an innovation might be overstating it”***
> > >
> > > * Understood, we would like to avoid any overstatement. We will rephrase the corresponding sections in abstract, introduction and conclusion. Specifically, we will say that “we leverage supervised attention [citations] to realize modularity in a complex robot manipulation task thereby enabling efficient transfer to new robots and scenarios.” We will make sure to cite the papers put forward by the reviewer. We sincerely thank you for those citations!
> > >
> > > **Responses to minor questions:**
> > > **- Line 233: Doesn’t the network predict target end-effector poses? Why is motion planning done towards a “target object”?**
> > > That line refers to the generation of training data, we will be revise the manuscript to make it clearer.
> > >
> > > **- How are supervision losses L_{k} obtained for each task from the demonstrations? How is supervised attention obtained in real-world experiments? Is it manually human-specified or automatically extracted from the demonstrations?**
> > > It is automatically extracted by projecting the object and end-effector position into the image frame. Two aspects however are manual, namely a.) a one-time robot-camera extrinsic calibration to get relative camera pose, and b.) for each module we define which embeddings it needs as input. More details in the attached pdf under Sec 2 and in particular in Sec 2.3.
> > >
> > > **- Could you better explain the “w/o Hierarchical Modularity” baseline? Does this baseline only train on the task loss and not on the intermediate attention losses? Or does it train on all the same losses as the full model, but jointly all at once without gradually introducing more complex tasks once simpler tasks have converged.**
> > > The latter (jointly all at once without gradual introduction). We will revise the manuscript to make this clear.
> > >
> > > **- Is this method of transfer across robots better than e.g. predicting grasp points / end-effector keypoints, and running a motion planner to control the robots?**
> > > It is more expressive in that it allows for arbitrary motions of the robot. Please see answer A4 above.
> > >
> > > **- Line 292: What are “register slots”? Were these defined anywhere in the paper? What kind of noise was added? Was this noise of a relative or absolute magnitude with respect to the distribution of values in these “register slots”?**
> > > As described in Line 172 of the paper, Register Slots are trainable tokens of embeddings that can propagate information throughout attention layers. Ultimately, they are used to store the output of a module so that it can be accessed as input in subsequent modules in the hierarchy. We realize this needs more explanation and have provided such in the attached pdf (Sec. 2).
> > >
> > > Thank you very much for the suggestions regarding phrasing etc. We will revise the manuscript accordingly!

---

> ### Author Response · Authors · 2022-08-21
> **Supplementary Materials**
>
> **Comment:**
>
> * PDF explaining the details of additional information
> * Video showing supplementary experiments
>
>
> **Zip File:**
>
> /attachment/114ce30a87bbee877a6160db8a6d5731aa394cda.zip

---

> ### Author Response · Authors · 2022-08-28
> **Check back**
>
> Dear Reviewer CVpY,
>
> We thank you for your valuable feedback! We made sure to address your feedback in the rebuttal above. In particular, a.) we will revise writing to avoid any overstatement, b.) we conducted a battery of new experiments that show different types of generalization capabilities, c.) the labeling process is largely automated and  human intervention/overhead boils down to doing a common camera calibration, d.) we clarify that we achieve a 9% - 20% improvement in overall task success over BC-Z (CoRL 2021) and completely outperform LP (NeurIPS 2020). In crafting our rebuttal, we went the extra-mile in reaching out to the authors of [1,3] to properly characterize their work and its capabilities (we can provide evidence of this, if requested). Most importantly, we created a new video containing the results of the new experiments, as well as a supplemental document providing more details as you suggested. We value your suggestions and feedback!

---

### Meta-Review · Area_Chair_GyuR · 2022-08-13

**Recommendation:** Accept (Poster)
**Confidence:** 5

**Metareview:**

# Strengths
- The paper describes an interesting approach to constructing policies
- Human interpretable components theoretically allow for better supervision and introspection of the policy

# Weaknesses
- The novelty should be clarified, e.g. although supervised attention is a promising concept it is not a novel contribution.
- The improved performance is rather small when considering the additional annotation burden
- The paper would benefit from some clarifications and additional details wrt the proposed approach

# Post-Rebuttal Update
The authors provided valuable clarifications in their responses to the reviewers that would improve the manuscript significantly.
I expect the authors to incorporate these clarifications and the additional experiments into the camera-ready version of the paper.
While I still think that the additional labeling is not negligible, especially in real-world demonstrations, I do see the added benefit.


**Best Paper Nomination:**

No

---

> ### Author Response · Authors · 2022-08-28
> **Response to Area Chair GyuR**
>
> Thank you for your acknowledgement of the strengths of our work, in particular that it is an interesting approach, and it allows for better supervision and introspection of the policy. We also thank you and reviewers for the feedback!
>
> In response to the reviewers' suggestions, we have performed a variety of additional new experiments that shed light on the generalization capabilities of our method. These experiments include investigations into the ability to deal with a.) occlusions, b.) unseen synonyms, c.) unseen object colors, d.) object scaling, e.) new objects, f.) unseen free-form instructions. Furthermore, as suggested by the reviewers, we added a whole new module for obstacle avoidance and show that it elegantly integrates into the system. All of these results can be found in the attached video and supplemental material. We have attached a video (in zip) which is an integral part of our rebuttal. We would therefore greatly appreciate it, if you could please watch the video.
>
> We are extremely grateful to reviewer fi3K who has already adjusted their score to “strong accept” as a result of our discussions in the rebuttal. Below our responses to previous comments in the meta-review:
>
> **C1:** ***The novelty should be clarified, e.g. although supervised attention is a promising concept it is not a novel contribution.***
>
> **A1:** As suggested by Reviewer CVpY, we will avoid overstatements. In detail, we will rephrase the corresponding sections in abstract, introduction and conclusion, and include explicitly “we leverage supervised attention [citations] to realize modularity in a complex robot manipulation task thereby enabling efficient transfer to new robots and scenarios.” We will make sure to cite the papers put forward by the reviewer. As highlighted in the title and in the body of the paper, the main focus of the paper is to achieve modularity in robotic systems through a clever use of attention.
>
> **C2:** ***The improved performance is rather small when considering the additional annotation burden.***
>
> **A2:** Labeling is largely automated as mentioned in response to Reviewer CVpY. Only for real-world experiments there is some human effort that needs to go towards robot-to-camera extrinsics calibration – a common task in robotics (see also supplemental material). Also, when compared to BC-Z (CoRL 2021) our method increases the task success by up to **20% in transfer tasks** (Fig. 5) and **9.3% even in tasks without transfer** (Tab. 2). Similarly, another SotA method LP (NeurIPS 2020) is not able to solve any of the tasks in this paper. Considering the complexity of our domain, which incorporates the combination of vision, language and control, we argue these are **significant margins**. We show in the robot transfer tasks that the method is data-efficient. The performance of our method using 160 demonstrations outperforms BC-Z with 320 demonstrations (Fig. 5), **cutting the data needed by more than half**. We were extremely careful in performing a fair and truthful comparison to other methods to **ensure reproducibility**. In light of the above results, we respectfully disagree with the statement that the improved performance upgrade is rather small.
>
> **C3:** ***The paper would benefit from some clarifications and additional details wrt the proposed approach.***
>
> **A3:** We totally agree with this comment. In the paper, we will provide concise descriptions of the inputs and outputs. We have also added a section in the supplementary material which describes the functionality of different supervision labels, as well as how we obtain them from demonstrations in a largely automated manner. More information in the supplementary material (Sec. 2.3.).